# *Withania somnifera* L.: Phenolic Compounds Composition and Biological Activity of Commercial Samples and Its Aqueous and Hydromethanolic Extracts

**DOI:** 10.3390/antiox12030550

**Published:** 2023-02-22

**Authors:** Milena Polumackanycz, Spyridon A. Petropoulos, Tomasz Śledziński, Elżbieta Goyke, Agnieszka Konopacka, Alina Plenis, Agnieszka Viapiana

**Affiliations:** 1Department of Analytical Chemistry, Medical University of Gdansk, 80-416 Gdansk, Poland; 2Department of Agriculture, Crop Production and Rural Environment, University of Thessaly, Fytokou Street, 38446 Volos, Greece; 3Department of Pharmaceutical Biochemistry, Medical University of Gdansk, 80-211 Gdansk, Poland; 4Department of Pharmaceutical Microbiology, Medical University of Gdansk, 80-416 Gdansk, Poland

**Keywords:** winter cherry, ashwagandha, phenolic profile, antioxidant capacity, antimicrobial activity, bioactive compounds, Solanaceae, gooseberry

## Abstract

In the present study, the chemical composition and bioactive properties of commercially available *Withania somnifera* samples were evaluated. The hydromethanolic and aqueous extracts of the tested samples were analyzed in terms of phenolic compound composition, ascorbic acid content, antioxidant and antibacterial activity, and acetylcholinesterase (AChE) and butyrylcholinesterase (BChE) inhibitory activities. Polyphenols and ascorbic acid content, as well as the antioxidant activity, were higher in the aqueous extracts than in the hydromethanolic extracts. Generally, aqueous extracts presented higher antioxidant activity than the hydromethanolic ones, especially in the case of 2,2-diphenyl-1-picryl-hydrazyl (DPPH) assay. Moreover, higher amounts of phenolic acids and flavonoids were found in the hydromethanolic extracts compared to the aqueous ones. Regarding the antibacterial properties, samples 4, 6, and 10 showed the best overall performance with growth-inhibitory activities against all the examined bacteria strains. Finally, the aqueous and hydromethanolic extracts were the most efficient extracts in terms of AChE and BChE inhibitory activities, respectively. In conclusion, our results indicate that *W. somnifera* possesses important bioactive properties which could be attributed to the high amounts of phenolic compounds. However, a great variability was recorded in commercially available products, suggesting significant differences in the origin of product and the processing method.

## 1. Introduction

*Withania somnifera* (L.) Dunal (Ashwagandha) belongs to the Solanaceae family and is also known as poison gooseberry, Indian ginseng, and winter cherry [1]. In Sanskrit, Ashwagandha means odor of the horse, originating from the aroma of the roots which resembles that of horse sweat, while the name “somnifera” in Latin means “sleep-inducer” which refers to its extensive use as an anti-stress remedy [2]. It originates in northwestern and central India and the Mediterranean region of North Africa and has been an important herb in Ayurveda and traditional medicine for over 3000 years [3,4]. It also occurs in Australia, Pakistan, Afghanistan, Jordan, Morocco, and Spain. In India, it is cultivated on a commercial scale in the states of Madhya Pradesh, Uttar Pradesh, Punjab, Gujarat, and Rajasthan [5]. The plant appears in WHO monographs on Selected Medicinal Plants and an American Herbal Pharmacopoeia monograph is also available [6,7].

*W. somnifera* is an important ingredient in many Ayurvedic formulations, which are currently commercialized in India and other countries in the world [5]. Most of the products of *W. somnifera* are used and sold as dietary supplements in the form of powder, syrup, infusions, ointments, tablets, and capsules [8]. In Ayurveda systems of medicine, the leaves, flowers, fruits, seeds, and roots of *W. somnifera* are also used for numerous therapeutic purposes [9]. Its leaves are bitter and have some medicinal uses against fever, as an anthelmintic agent, and for painful swelling. The flowers are depurative, astringent, diuretic, and aphrodisiac, while the fruits have been used in the treatment of tumors and tubercular glands, carbuncles, and skin ulcers [10,11]. Moreover, the fruits are also used as a coagulant agent in curdling plant milk to prepare vegetarian cheese [12]. The seeds of *W. somnifera* are anti-helminthic, remove white spots from the cornea, and increase sperm count and testicular growth [13]. Among the various plant parts of *W. somnifera* which have been claimed to have a large variety of health-promoting effects, the roots are the most popular. Their powder and preparations are consumed extensively as a functional food for promoting vitality and virility. They are used to prepare a tonic which revitalizes the body, promotes longevity, augments defense against infectious diseases, and arrests the aging process [14,15]. According to ethnobotany studies, roots and leaves of the species are also used as a hypnotic in alcoholism and emphysematous dyspnea [16,17]. Moreover, *W. somnifera* has other medicinal uses including immunomodulatory [18], antidiabetic and neuroprotective [5], anticancer [10], and anti-inflammatory [19]. Additionally, it is also useful as antibiotic, antioxidant, deobstruent, aphrodisiac, diuretic, and sedative [5,20].

Phytochemical investigations on *W. somnifera* have revealed the presence of various chemical constituents such as steroidal compounds, alkaloids, phenolic compounds, saponins containing an additional acyl group, withanolides with a glucose at carbon 27, etc. [21,22,23,24]. To date, more than 12 alkaloids, around 40 withanolides, and several sitoindosides are reported from the aerial parts, roots, and berries of *W. somnifera* [24]. The leaves are reported to contain five unidentified alkaloids (yield, 0.09%), 12 withanolides, many free amino acids, chlorogenic acid, glycosides, glucose, condensed tannins, and flavonoids [25]. Moreover, the roots contain alkaloids, amino acids, steroids, volatile oil, starch, reducing sugars, glycosides, hentriacontane, dulcitol, and withaniol [26]. Out of these metabolites, withanolides and phenolic compounds are mainly credited with the broadly acclaimed curative properties of *W. somnifera* [27]. Withanolides stimulate activation of immune system cells, while phenolic compounds are closely associated with the antioxidant activity of the plant [28]. Moreover, these compounds have also shown antiviral activity [29], with distinct effects on the viral receptor, which might be potent against COVID-19 [30,31].

There are many reports on the withanolides and alkaloids, but only a few studies are available regarding the phenolic composition and antioxidant activity of *W. somnifera* [32,33,34]. For this reason, the study of phenolic compounds is important both for their characterization and to facilitate more efficient use of this important plant resource. Therefore, the aim of the present study was to assess the phenolic composition of infusions and hydromethanolic extracts of commercial samples of *W. somnifera* and further evaluate their antioxidant, antibacterial, acetylcholinesterase (AChE), and butyrylcholinesterase (BChE) inhibitory activities. The results of this study could provide new important insights into the contribution of phenolic compounds to the bioactive properties of *W. somnifera*, focusing on over-the-counter commercially available products, while they also assess potential differences between the different types of products in terms of antioxidant activities. Individual phenolic acids and flavonoids were quantified in hydromethanolic and aqueous extracts obtained from the samples under study; total phenolic compounds, total flavonoids, total phenolic acids, and L(+)-ascorbic acid content were also determined. The antioxidant activities of the tested extracts were evaluated using DPPH and ABTS free radical scavenging assays, as well as ferric-reducing/antioxidant power (FRAP) assay, while antibacterial activity was determined only for aqueous extracts of *W. somnifera* specifically because of their widespread use as tea infusions. Finally, AChE and BChE inhibitory activity was determined for both the hydromethanolic and aqueous extracts.

## 2. Materials and Methods

### 2.1. Plant Material

Eighteen commercial samples (no 1–18) of *Withania somnifera* L. (Dunal) were purchased in a local supermarket (Auchan), herbal store (Fragaria, Eko-Ziola, Nagietek), and pharmacy store (Gemini) in Gdansk, Poland. The majority of samples came from India and contained roots of *W. somnifera*. Only samples no 6 and 9 contained rhizome tissues and whole plant, respectively. Eight samples (no 1, 3, 4, 5, 6, 8, 14, and 17) were in capsule form, two samples (no 2 and 9) were in tablet form, and eight samples (no 7, 10, 11, 12, 13, 15, 16 and 18) were in powder form (Table 1). All samples were pulverized in a water-cooled Knifetec 1095 grinder (Foss Tecator, Höganäs, Sweden) and the homogenized samples were stored in a light-proof desiccator until further analysis.

### 2.2. Reagents and Standard Solutions

For the chemical analyses, the following high-purity standards (>98%), e.g., 2,2-diphenyl-1-picrylhydrazyl (DPPH reagent), 2,20-azinobis(3-ethylbenzothiazoline-6-sulfonic acid) diammonium salt (ABTS reagent), 4-chloro-7-nitrobenzofurazan (NBD-Cl), and ten chemical standards: gallic acid (GA), catechin (CAT), vanillic acid (VA), caffeic acid (CA), *p*-coumaric acid (*p*CA), ferulic acid (FA), sinapic acid (SYN), rutin (RUT), quercetin (Q), and naringenin (NAR), were purchased from Sigma-Aldrich (St. Louis, MO, USA) [35]. Aluminum chloride (AlCl_3_) was obtained from Across Organics (Morris Plains, NJ, USA) and HPLC-grade acetonitrile (ACN) from Avantor (Central Valley, PA, USA) [35], while the rest of the reagents were obtained from POCh (Gliwice, Poland) [35]. The redistilled water was prepared by triple distillation of water in a Destmat^®^ Bi-18 system (Heraeus Quarzglas, Hanau, Germany), as already described by Polumackanycz et al. [35].

The determination of total phenolic compounds (TPC), total flavonoids (TF), total phenolic acids (TPA), L(+)-ascorbic acid (ASA) contents, and antioxidant activity was performed with a Metertech UV/Vis spectrophotometer (Nankang, Taiwan) by measuring the absorbance with 10 mm quartz cuvette at the wavelength described below in the corresponding sections [35].

### 2.3. Sample Preparation

The hydromethanolic extracts were prepared by sonicating 1.0 g of each sample with 7 mL of methanol–water mixture (80:20; *v/v*) for 20 min at 20 °C using an ultrasonic bath (Emag, Salach, Germany) [35]. Then, the extract was centrifuged in an EBA-20S centrifuge (Hettich, Tuttlingen, Germany) for 10 min at 80,000 rpm and the supernatant was transferred into a 20 mL volumetric flask [35]. The extraction process was performed twice and the obtained supernatant were put together in the same flask and diluted up to 20 mL using a mixture of methanol and water (80:20; *v/v*) [35].

The infusions (aqueous extracts) were prepared by adding 1.0 g of sample into 100 mL of boiling distilled water, then left to stand for 10 min at room temperature, and finally filtered with a Whatman filter paper no. 113 (Sigma-Aldrich, St. Louis, MO, USA) [35].

The obtained extracts (both hydromethanolic and aqueous ones) were passed through a 0.25 μm nylon filter film (Mecherey, Nagel, Germany) and then 20 μL of the filtrate were injected into the HPLC system [35].

### 2.4. Phytochemical Composition

#### 2.4.1. Chromatographic Conditions

Chromatographic separation of phenolic compounds was performed according to the protocol previously described by Viapiana et al. [36]. The equipment used was a Merck-Hitachi LaChrome device (Darmstadt, Germany) equipped with an L-7420 UV-Vis detector, L-7200 autosampler, L-7360 thermostat and D-7000 HPLC System Manager, ver. 3.1 (Merck-Hitachi, Darmstadt, Germany) [36]. The detection and quantification were performed using the method described previously [36]. The detection wavelengths were set at 280 nm (GA, VA, CAT, NAR), 320 nm (CA, FA, *p*CA, SYN), and 370 nm (RUT, Q) [36].

Phenolic compound identification was carried out by comparing the retention times of the detected compounds with those of commercial standards, as well as by spiking a sample with commercial standards. The limit of detection (LOD) and limit of quantification (LOQ) are presented in Table 2.

#### 2.4.2. Total Phenolic Compounds (TPC), Flavonoid (TF), and Phenolic Acid (TPA) Contents

For total phenolic compound determination, the Folin–Ciocalteu [37] method was implemented and the results were presented as mg of gallic acid equivalents per gram of dry weight of sample (mg GAE/g DW). The total flavonoid content was determined according to the European Pharmacopoeia [38] and the results were presented as mg of quercetin equivalents per gram of dry weight of sample (mg QE/g DW).

Finally, total phenolic acid content was evaluated using Arnov’s reagent according to the procedure described in the Polish Pharmacopoeia VI [39]. The results were presented as mg of caffeic acid equivalent per gram of dry weight of sample (mg CAE/g DW).

#### 2.4.3. L(+) Ascorbic Acid Content

Ascorbic acid was determined based on the protocol previously described by Abdelmageed [40] after slight modifications which are reported by Viapiana et al. [35]. The results were presented as mg of ascorbic acid per gram of dry weight of sample (mg ASA/g DW).

### 2.5. Bioactive Properties

#### 2.5.1. Antioxidant Activity

The antioxidant activity was determined using four different assays, namely, the 2,2-diphenyl-1-picryl-hydrazyl (DPPH) assay [41]; the 2,2′-azinobis-(3-ethylbenzothiazoline-6-sulfonate) (ABTS) assay [42], with slight modifications which are reported by Viapiana et al. [35]; and the ferric-reducing antioxidant power (FRAP) assay [43].

#### 2.5.2. Antibacterial Activity

The antimicrobial activity analysis of infusions (aqueous extracts) was performed according to the protocols of EUCAST (European Committee on Antimicrobial Susceptibility Testing) and CLSI (Institute of Clinical and Laboratory Standards), as previously described by Polumackanycz et al. [44].

##### Bacterial Strains and Growth Conditions

Antibacterial activity was tested according to the method previously described by Polumackanycz et al. [44], using the following bacteria: Gram-positive strains, e.g., *Staphylococcus aureus* ATCC 6538, MRSA (18582, 6347, N315, 12673). *Staphylococcus epidermidis* ATCC 14990, *Bacillus subtilis* ATCC 6633, *Corynebacterium diphtheriae*, group A β-hemolytic *Streptococcus,* and *Streptococcus pneumoniae* (from the own collection of the Department of Pharmaceutical Microbiology), and Gram-negative strains: *Escherichia coli* ATCC 8739, *Pseudomonas aeruginosa* ATCC 9027.

##### Agar Well Diffusion Assay

The agar well diffusion test was used as a preliminary study of the antimicrobial activity, using the following strains *Staphylococcus aureus* ATCC 6538, *Escherichia coli* ATCC 8739, and *Pseudomonas aeruginosa* ATCC 9027, according to the methodology previously described by Polumackanycz et al. [44].

##### MIC and MBC Assays

The Minimal Inhibitory Concentration (MIC) and Minimal Bactericidal Concentration (MBC) were tested using the broth microdilution technique. Due to the fact that the ashwagandha extracts were colorful, determination of the MIC values was troublesome. All antimicrobial activity assays were performed in triplicate for at least two independent experiments.

#### 2.5.3. AChE and BChE Inhibitory Activity

The inhibitory activities of AChE and BChE were assayed by Ellman’s method on a 96-well microplate similar to that previously described in the literature [45].

### 2.6. Statistical Analysis

For each sample, three extracts were prepared, while each extract was analyzed in triplicate. The obtained results were presented as mean values and standard deviation (SD). The statistical analysis was performed using the two-way analysis of variance (ANOVA) followed by means comparison with Tukey’s HSD test (*p* < 0.05) and Student’s *t* test (*p* < 0.05) for the means of the same extraction method and the same sample, respectively. Pearson’s correlation analysis was used to reveal the relationship between *W. somnifera* extracts and the detected chemical profile and biological activities. For the analysis of data, the Statistica 10 software (StatSoft Inc., Tulsa, OK, USA) was used. Finally, a principal component analysis (PCA) was carried out to evaluate the contribution of each variable to the total diversity aiming to classify the studied samples based on their chemical profile, the bioactive properties, and the extraction protocol. For this analysis, the statistical software Statgraphics 5.1.plus (Statpoint Technologies, Inc., Warrenton, VA, USA) was implemented.

## 3. Results and Discussion

### 3.1. Validation Method for the Identification of Phenolic Compounds

The validation of the applied method was performed by evaluating the following parameters: linearity, limits of detection (LOD) and quantification (LOQ), intra- and inter-day precision, recovery, and stability. In Table 2, good linearity was found over the determined ranges for all the detected compounds, with correlation coefficient (r) values significantly higher than 0.985. The LOD and LOQ were calculated in accordance with the following equations: LOD = 3.3Sxy/b and LOQ = 10Sxy/b, where Sxy is the standard deviation of the response and b is the slope of the calibration curve. The values of LODs and LOQs were less than 3.5 and 13.2 μg/mL, respectively. These results show that the analytical method had excellent resolution and sensitivity. Intra-day precision was validated with a standard solution of assayed phenolic compounds three times within one day, while inter-day precision was validated with the same standard solution over three consecutive days. Consequently, the precision was acceptable, and the coefficient of variation values ranged from 0.6% to 1.4% and from 1.2% to 2.5% for intra- and inter-day variations, respectively.

The mean recovery was also found to be in a satisfactory range, e.g., 93.1–97.8%, with a relative standard deviation (RSD) less than 3.5%. The peak areas and retention times of the detected phenolic compounds were analyzed every eight hours within 48 h for the stability test and they were found to be quite stable, while retention CV was lower than 1.9% for peak area and 0.4% for retention time.

### 3.2. Analysis of TPC, TF, TPA and ASA

The results of total phenolic compounds (TPC), total flavonoids (TF), total phenolic acids (TPA), and L(+)-ascorbic acid (ASA) content in hydromethanolic and aqueous are shown in Table 3. The obtained results revealed that TPC, TF, TPA, and ASA content in the aqueous extracts were significantly higher (*p* < 0.05) than those in the hydromethanolic extracts of *W. somnifera*. In contrast to our study, Alam et al. [34] analyzed hydromethanolic extracts of *W. somnifera* roots and also found higher values of TPC and TF at levels of 17.80 mg GAE/g DW and 15.149 mg CEQ/g DW, respectively. Similar results for TPC were obtained by Dhanani et al. [46] for aqueous extracts of *W. somnifera* roots (14.90–18.72 mg GA/g DW). Moreover, higher contents for TPC and TF were obtained by Ganguly et al. [47] for hydromethanolic (97.38 µg GA/mg of extract and 63.49 µg Q/mg extracts, respectively) and aqueous extracts (53.9 µg GA/mg of extract and 38.66 µg Q/mg of extracts, respectively) in *W. somnifera* roots, while similar levels of TPC were reported by Yadav and Rai [48] for methanolic extracts of *W. somnifera* roots (52.811 mg GAE/100 g DW). The reported values are higher than those obtained in this study. An explanation for the variability detected among the phenolic contents in ashwagandha samples could be due to the different extraction procedures and analytical methods used in each work [49,50]. In addition, it has been suggested that phenolic compounds in plants vary according to growing conditions such as drought, temperature changes, pollution, UV light, and pathogen attacks, among others, or the genotype tested [51,52,53]. Moreover, the expression of the results could also explain these contradictory findings, as in the case of Chaudhary et al. [54] who expressed the results of TPC and TF of aqueous extracts of *W. somnifera* roots as % (24.70 and 1.83, respectively), while Paul et al. [55], who tested the methanolic extracts of *W. somnifera* roots, expressed TPC and TF contents as mg GAE/mL (0.39–0.52) and mg QE/mL (0.50), respectively. In particular, Chaudhary et al. [54] suggested that natural matrices contain complex mixtures of polyphenols which may reveal different bioactive properties depending on the amounts of the active components; thus, the higher amounts of total polyphenols are not always accompanied by higher antioxidant activities. On the other hand, Paul et al. [55] who tested two different *W. somnifera* samples (one indigenous and one imported root sample), suggested significant differences in terms of total phenolic compounds and total flavonoids. Therefore, direct comparison of the results from different reports is not always possible. Finally, these contradictions with the literature reports could be due to the fact that in other studies, raw material was used for the analyses, whereas in our study, all the tested samples were already processed in capsule, tablet, or powder form without any details of processing method available [56]. To the best of the authors’ knowledge, this is the first report of the TPA and ASA in *W. somnifera* aqueous and hydroalcoholic extracts.

### 3.3. Evaluation of Antioxidant Activity

The antioxidant potential of the tested *W. somnifera* commercial samples was evaluated by the use of DPPH, ABTS, and FRAP tests, and the results are shown in Table 3. Generally, for the three methods, aqueous extracts of *W. somnifera* presented higher antioxidant activity than the hydromethanolic ones. The superior antioxidant activity obtained for aqueous extracts could be explained by their higher content of phenolic compounds, since phenolic compound content has been associated with high antioxidant activity in various species [57,58,59]. Antioxidant activity results for hydromethanolic extracts ranged from 19.48 to 91.17 mg TE/100 g DW by DPPH assay, 16.54–124.84 mg TE/g DW by ABTS assay, and 3.87–45.57 µmol Fe^2+^/g DW by FRAP method. On the other hand, for aqueous extracts, DPPH, ABTS, and FRAP values ranged from 93.66 to 281.92 mg TE/100 g DW, 21.31 to 90.04 mg TE/g DW and 13.17 to 40.01 µmol Fe^2+^/g DW, respectively. Sample no 6 was characterized by the highest values of ABTS (above 124 mg TE/g DW in hydromethanolic extracts and above 90 mg TE/g DW in aqueous ones), while sample no 9 showed the highest FRAP values (above 84 µmol Fe^2+^/g DW in hydromethanolic extracts and above 40 µmol Fe^2+^/g DW in aqueous extracts). Moreover, infusion of sample no 13 was characterized by the lowest ABTS and FRAP values (below 7 mg TE/g DW and 14 µmol Fe^2+^/g DW, respectively). Finally, the hydromethanolic extract of sample no 1 showed the lowest DPPH values (below 20 mg TE/100 g DW), while its aqueous extract was characterized by the highest DPPH values (above 280 mg TE/100 g DW).

The DPPH and ABTS values of *W. somnifera* extracts obtained in this study cannot be compared with those in literature reports due to the difference in calculation units. In the literature, the antioxidant activity for *W. somnifera* extracts was expressed as the percentage of inhibition and the half-maximal inhibitory concentration (IC_50_). For example, Dhanami et al. [46] determined the antioxidant activity in aqueous extracts of *W. somnifera* roots and obtained DPPH value in the range of 0.40 to 1.20 mg/mL (IC_50_) and ABTS values in the range of 2.14 to 2.68 mg/mL (IC_50_). On the other hand, Chaudhary et al. [54] reported higher DPPH and ABTS values (4612.17 and 541.76 µg/mL, IC_50_), while Ganguly et al. [47] obtained lower values of DPPH (111.31µg/mL, IC_50_). Furthermore, in aqueous extracts of *W. somnifera* roots, Chaudhary et al. [54] recorded a FRAP value at the level of 13.61 mmol ascorbic acid/mL, and Yadav and Rai [48] determined ABTS value at the level of 19.54% of inhibition. Similarly, for hydromethanolic extracts of *W. somnifera* roots Ganguly et al. [47] obtained DPPH value higher than 30 µg/mL (IC_50_), while Alam et al. [34] reported DPPH value at the level of 59.16% of inhibition. All these results from the literature indicate a significant variation in the antioxidant activity of *W. somnifera* depending on the means of extraction, while the extraction protocol may also affect the antioxidant potential of natural matrices [60]. Moreover, according to Nile et al. [61], the extraction condition (e.g., solvent, extraction time, and extraction temperature) may affect the antioxidant potential of dried leaves and roots of *W. somnifera* due to differences in withanoside and withanolide content. In the study of Simur [61], where methanol, acetone, and hexane extracts of *W. somnifera* leaves were tested, a great variation in antioxidant potential was also reported. Another aspect to be considered is the growth stage of plants at harvesting, since according to Fernando et al. [62], harvesting after flowering may increase the total antioxidant activity of the extracts from different plant parts of the species.

### 3.4. Analysis of Individual Phenolic Compounds

Table 4 presents a profile of individual phenolic compounds in the tested extracts of commercial *W. somnifera* samples. In general, higher amounts of phenolic acids and flavonoids were found in the hydromethanolic than in the aqueous extracts. In the hydromethanolic extracts, only five phenolic compounds (GA, CAT, Q, CA and RUT) were found in all analyzed samples. SYN was determined only in four samples, *p*CA in six samples, VA in eight samples, NAR in twelve samples, and FA in eleven samples. Moreover, CAT and Q were more abundant in the hydromethanolic extracts, 1.96 and 1.35 mg/g DW, respectively, while SYN and VA were determined in the lowest concentration (135 and 108.99 µg/g DW, respectively). In the case of aqueous extracts, only CAT was determined in all samples, and together with GA and Q, was found in the highest amounts (7.65, 1.05, and 0.81 mg/g DW, respectively). SYN was determined only in three samples and was found in the lowest concentration. In addition, the hydromethanolic extracts of samples no 4, 5, and 8 were the richest in phenolic compounds, especially in CAT and RUT, while samples 10 and 18 had the lowest content. For aqueous extracts, samples no 2 and 6 were the richest in phenolic compounds, while samples no 12 and 13 were the poorest. Moreover, in samples no 12 and 13, only RUT and VA were found.

The literature data on phenolic compound composition in *W. somnifera* roots are scarce. Alam et al. [34] found six times higher CAT content (12.82 mg/g DW) in hydromethanolic extracts of *W. somnifera* roots than in this study, while GA, SYN, VA, *p*CA, and NAR were not detected. The content of CA and FA in methanolic extracts of *W. somnifera* roots reported by Tomar et al. [33] was lower than in this study. However, these authors presented their results on a fresh weight basis, while the samples were obtained from cultivated plants. These variations in the literature could be due to either intrinsic and/or extrinsic factors, such as cultivation practices, storage conditions, type of soil, climatic factors, and technological treatments [45,46,63]. Moreover, plant part is crucial for the phenolic composition of the obtained extracts, as already reported by Tomar et al. [33], who detected significant differences among plant parts (stems, roots, and leaves). Similarly to our study, significant differences in phenolic compounds composition were reported by Alam et al. [34] and Tomar et al. [33], who tested hydromethanolic and methanolic and chloroform extracts, respectively. In particular, Alam et al. [34] detected CAT (19.48 mg/g DW) and NAR (0.50 mg/g DW), while GA, VA, and *p*CA were not detected in the hydromethanolic extracts of *W. somnifera* fruits. On the other hand, Tomar et al. [33] found 1.5 µg/g FW of CA in methanolic extracts of *W. somnifera* leaves, while Alam et al. [34] detected GA, SYN, VA and *p*CA (28.38; 0.18; 0.30; 0.15 and 0.80 mg/g DW, respectively) and found no detectable amounts of NAR in the hydromethanolic extracts of *W. somnifera* leaves.

### 3.5. Evaluation of Antibacterial Activity

The antibacterial tests were carried out using the agar diffusion method. Antimicrobial activity analyses (assessed in terms of inhibition zone) of *W. somnifera* aqueous extracts, tested against *S. aureus* ATCC 6538, *E. coli* ATCC 8739, and *P. aeruginosa* ATCC 9027 were performed and only ten of the tested extracts showed significant activity against the three selected bacteria strains. Then, MIC and MBC values of *W. somnifera* extracts that showed antibacterial activity in diffusion assay were further evaluated (Table 5). The range of MIC and MBC values of extracts was 0.25–32 and 1–32 mg/mL, respectively. The lowest MIC value (0.25 mg/mL) was recorded against *S. pyogenes* (samples no. 1, 3, 4, and 6) and *C. diphtheria* (sample no. 3). Moreover, *S. pyogenes* was the most sensitive of the examined bacterial strains with MIC values ranging between 0.25 and 4 mg/mL, whereas *E. hirae*, *E. faecalis,* and MRSA 12,673 were the most resistant to tested *W. somnifera* extracts. Furthermore, extracts prepared from samples 4, 6, 10, and 15 showed the best overall performance with growth-inhibitory activities against all the examined bacteria strains, while sample no 3 was characterized by the lowest antibacterial activity compared to other samples, except for the case of MIC values of aqueous extracts against *S. pyogenes* and *C. diphtheria*. Moreover, MBC values were much higher compared to the MIC values, thereby confirming that *W. somnifera* extracts may have bactericidal effects at high concentrations and bacteriostatic effects at lower ones.

Our results are in agreement with literature reports where the antibacterial activity of *W. somnifera* extracts is indicated. For example, the methanol, ethanol, aqueous, and butanol root extracts of the species were effective in the inhibition of multidrug-resistant *Staphylococcus aureus* strains tested by the agar well diffusion method, although a significant variation between the different extracts was recorded (butanol extracts were the most effective, whereas the aqueous ones the least effective) [64]. The differences between the various extracts could be associated with the presence of specific active components, as in the case of total phenolic compounds, total flavonoids and total phenolic acids, which were the highest in sample no 6 that were effective against most of the studied bacteria. On the other hand, this was not the case with samples 4, 10, and 15 which showed a variable content of phenolic compounds, indicating that other compounds may also contribute to the bioactive properties of *W. somnifera* extracts and their antimicrobial effects in particular. Similarly, the methanol and hexane extracts of *W. somnifera* leaves and roots were effective against *Salmonella typhimurium* and *Escherichia coli* [65], while according to Rizwana et al. [66] the acetonic extracts of roots were more effective against *Klebsiella pneumoniae* and methicillin-resistant *Staphylococcus aureus* (MRSA) than the methanolic and ethanolic ones. Moreover, the methanolic leaf extracts were very effective against several bacteria present in pus samples of patients, including methicillin-resistant *Staphylococcus aureus* and *Enterococcus* spp., while Mehrota et al. [67] reported that aqueous extracts of roots were effective against the same pathogen. Murugan et al. [68] and Gebeyehu et al. [69] suggested withaferin A as the main active component in leaf extracts against *Pseudomonas aeruginosa*, *S. aureus*, *Streptococcus pneumoniae*, *Salmonella typhi*, and *E. coli*; while Ha et al. [70] reported withanolide glycosides as the main bioactive compounds of roots. On the other hand, according to Balkrishna et al. [71], fatty acids were responsible for the antibacterial effects of fixed seed oils on *Salmonella enterica*. Similarly to our study, Mehta et al. [72] suggested significant differences in the efficacy of methanolic and ethyl acetate extracts of leaves against *Salmonela typhimurium* strains, which highlights the impact of the extraction protocol on the bioactive properties of the extracts. Moreover, each plant part seems to possess specific antibacterial properties which could be explained by the difference in its chemical profile and the abundance of specific bioactive components.

### 3.6. AChE and BChE Inhibition Results

The acetylcholinesterase (AChE) and butyrylocholinesterase (BChE) inhibitory activity of the hydromethanolic and aqueous extracts of *W. somnifera* L. is presented in Table 6.

The inhibitory activities of the studied samples showed great variation for both enzymes. In particular, for the aqueous extracts, the highest IA for AChE was 36% of the activity of the control sample (sample 14), whereas samples 7, 9, and 17 showed no inhibition of this enzyme. In the case of BChE, the highest IA among the aqueous extracts was recorded for sample 13 (34% of the activity of the control sample), and samples 1, 6, and 14 did not have any significant activity, whereas for sample 7 a slight increase in the BChE activity was detected. Among the hydromethanolic extracts, the highest IA for AChE was found for sample 8 (21% of the activity of the control sample), whereas samples 5, 7, and 17 did not inhibit the activity of the enzyme. Similarly, for BChE, the highest IA was recorded for sample 10 (16% of the activity of the control sample), while all the tested hydromethanolic extracts showed inhibition activity against BChE. When comparing the mean IA of AchE for all the aqueous extracts with the respective mean of hydromethanolic extracts, the former recorded a lower IA (64% comparing to 76% for hydromethanolic extracts), whereas for BchE, the hydromethanolic extracts had lower IA (40% comparing to 78% for aqueous extracts). The inhibition of AChE and BChE by a known inhibitor of both enzymes (9-Amino-1,2,3,4-tetrahydroacridine hydrochloride hydrate) that served as a positive control is presented in Appendix A.

The inhibition activity of *W. somnifera* against AChE has been previously reported in the literature, indicating the positive health effects for the treatment of neurodegenerative diseases and memory-related disorders such as Alzheimer’s disease [73,74,75,76,77]. For example, Pai et al. [78] suggested the positive effects of hydroalcoholic extracts and withanolide-A in in vitro and in silico inhibition of AChE, while Khan et al. [74] and Choudhary et al. [79] also highlighted the importance of withanolides to these particular properties. An earlier in vivo study with adult male Wistar rats also identified withanolide-A as the main bioactive compound responsible for the memory-improving effects of *W. somnifera* [80], while more recent reports suggested the positive effects of withanone obtained from leaf and root extracts against cognitive dysfunction [81,82]. Similarly, Singh et al. [83], who isolated nine distinct withanolides from ethanolic fractions of *W. somnifera* roots, identified five potent compounds with high AChE inhibitory activity, with 12-deoxywithastramonolide being the most efficient withanolide. The ethanolic extracts of roots were also effective against both AChE and BChE [84]. On the other hand, Raza et al. [85] who tested the AChE and BChE inhibitory activity from *W. somnifera* extracts (hexane, ethyl-acetate, butanol), recorded higher efficiency for the ethyl-acetate extracts for both enzymes and further suggested that these properties could be attributed to the higher recovery of phenolic compounds for these particular extracts. Moreover, the recent study of Tousif et al. [86] identified more than 100 metabolites in methanolic extracts of *W. somnifera* leaves fractionated with chloroform which was the most effective against Ache and BChE, including phenolic compounds, flavonoids, lignins, limonoids, steroids, terpenoids, and withanolides. This finding is in agreement with the report of Maya and Sarada [87] who suggested important AChE inhibitory activity of methanolic extracts of roots, despite its low content of polyphenols. Apart from single-extract treatments, Khan et al. [88] recently suggested the synergistic effects of the aqueous extracts of *W. somnifera* and *Myristica fragrans* which significantly increased the IC_50_ values in in vitro anticholinesterase assays, compared to treatments with single extracts.

### 3.7. Correlation Analysis

Pearson’s correlation analysis was implemented to determine the relationship between the detected phenolic compounds and the antioxidant activity of the *W. somnifera* samples (Appendix A). The correlation analysis showed 48 statistically significant correlations (*p* < 0.05) for the hydromethanolic extracts. The highest correlations were obtained between TPC-FRAP (0.875), TPC-TPA (0.866), and TPA-SYN (0.850). Moreover, the correlations between DPPH and TF, ASA, GA, NAR, *p*CA, and RUT were negative (−0.487; −0.705; −0.490; −0.573; −0.630 and −0.547, respectively), indicating that the specific compounds not only do not contribute to the antioxidant potential of the tested extracts but also that they may have a negative effect regarding the activity determined via the DPPH assay. On the other hand, 38 statistically significant correlations were found in the aqueous extracts, while the highest correlation (0.824) was obtained between RUT and TPA. In addition, the correlations of antioxidant activities and TPA, GA, SYN, and *p*CA in the aqueous extracts were found to be moderately positive, while the relationship between FRAP and Q was moderately negative (−0.615). Finally, AChE was significantly correlated with VA (−0.476) in the aqueous extracts. The obtained results suggest the crucial role of specific phenolic compounds as antioxidant constituents in *W. somnifera*, thus contributing to the overall bioactive properties of the tested commercial samples. Moreover, those cases where values of Pearson’s correlation coefficients were below 0.45 suggest that the specific constituents that occurred separately in the tested extracts could not be responsible for the antioxidant properties determined with the respective assays.

### 3.8. Principal Components Analysis

Principal component analysis (PCA) is widely used when aiming to reduce the complexity of multivariate data, as well as to identify patterns and express data in ways that highlight similarities and differences among the tested treatments. In our study, the aim was to identify groups of samples with similarities in terms of phenolic compound composition and antioxidant activity, as well as depending on the extraction protocol. The first five principal components (PCs) were associated with Eigen values higher than 1 and explained 79.93% of the cumulative variance, with PC1 accounting for 39.7%, PC2 for 18.2%, PC3 for 8.5%, PC4 for 8.0%, and PC5 for 5.4%. PC1 was positively correlated with TF, ASA, CAT, and DPPH, and negatively correlated with GA, NAR, RUT, and AChe. PC2 was positively correlated to Q and negatively correlated to TPA, FA, CAT, *p*CA, ABTS, and FRAP. Finally, PC3 was positively correlated to TPA, VA, RUT, FA, NAR, and CA, and negatively correlated to Q and SYN. These results indicate the correct implementation of the PCA, allowing differentiation between the tested samples depending on the extraction protocol, as shown in the corresponding scatterplots and loading plots (Figure 1, Figure 2 and Figure 3). The scatterplot (Figure 1) shows a clear discrimination of the tested samples according to the extraction protocol where all the aqueous extracts form a distinct group. PC1 discriminates sample 4 due to high content of CAT and TF and sample 6 due to high content of ASA. PC2 discriminates sample 1 due to low content of TPA and low values of ABTS, sample 6 due to low content of FA, sample 8 due to high content of Q and low content of SYN. Finally, PC3 discriminates sample 1 due to high contents of RUT and NAR, sample 2 due to high contents of FA and NAR, sample 4 due to high content of FA, sample 6 due to high content of TPA, and sample 8 due to high content of CA and TPA.

Moreover, the loading plot of the first two components also revealed groups of positively correlated variables (Figure 2). In particular, the upper left quadrant includes Q and AChe; the lower left quadrant comprises RUT, NAR, CA, GA, FA, SYN, and *p*CA; the upper right quadrant includes DPPH, TF, and BChe; and the lower right quadrant includes CAT, VA, ASA, TPC, TPA, FRAP, and ABTS. Similarly, the loading plot of PC1 and PC3 (Figure 3) revealed groups of positively correlated variables, namely, the upper left quadrant includes Ache, FA, NAR, CA, and RUT; the lower left quadrant comprises GA, Q, SYN, and *p*CA; the upper right quadrant includes BChe, VA, TPA, TPC, CAT, and FRAP; and the lower right quadrant includes ASA, ABTS, TF, and DPPH.

## 4. Conclusions

Natural products are promising sources of bioactive compounds and have been widely used in folk and traditional medicine, while they may provide important compounds for the design of novel drugs and medicines. The current market trends for functional foods and nutraceutical products has created a market niche for over-the-counter products which claim health effects and well-being improvement. However, considering the variability of the composition of raw materials, there is an urgent need to evaluate the chemical composition and the bioactive properties of commercially available formulations of natural products. In this context, our results indicate a significant variability in the chemical profile, and the antioxidant and antibacterial properties among the tested commercial samples of *Withania somnifera* associated with differences in chemical composition of the formulations due to the varied origin of the raw materials and the plant part used, as well the extraction protocol (hydromethanolic and aqueous extracts) implemented in our study. In particular, the aqueous extracts were more abundant in polyphenols with catechin and quercetin being found in the higher amount in both extracts, whereas synapic and vanillic acids were found in the lowest content. Moreover, aqueous extracts were richer in ascorbic acid and recorded higher antioxidant activity (especially in the case of DPPH assay) than the hydromethanolic extracts. Similarly, the aqueous extracts recorded higher antibacterial efficiency than the hydromethanolic ones, while roots in capsule form from India (e.g., samples 4 and 6), or ground roots from unknown origin or from India (e.g., samples 10 and 15, respectively) showed the best overall performance. On the other hand, the hydromethanolic extracts showed higher inhibitory activity against AChe than the aqueous extracts, whereas the opposite trend was recorded for the BChe activity. Finally, phenolic compounds were significantly positively correlated with the recorded biological activity of the tested extracts, indicating their important antioxidant potential in *W. somnifera* extracts. In conclusion, although *W. somnifera* is associated with several beneficial health effects, the products available over the counter do not ensure that they always possess these effects and further control is needed. Moreover, the processing of the studied products highlighted the importance of the extraction method to the obtained bioactive compounds and by extension to the potential health effects.

## Figures and Tables

**Figure 1 antioxidants-12-00550-f001:**
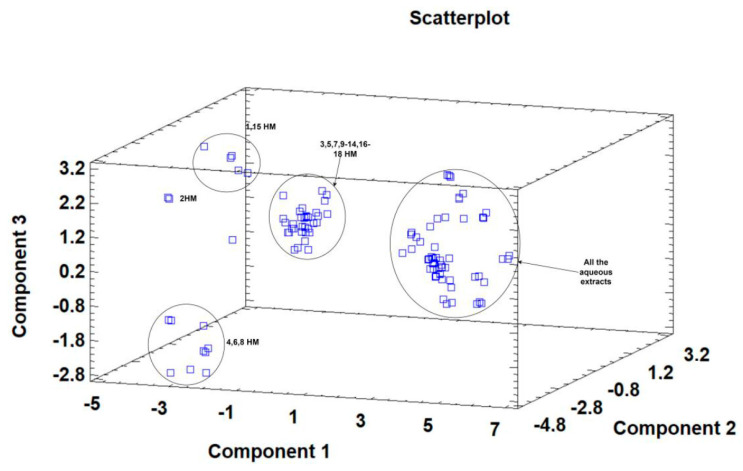
Three-dimensional scatterplot of principal components 1, 2, and 3 for the tested *Withania somnifera* samples extracted with two different protocols (hydromethanolic and aqueous extracts).

**Figure 2 antioxidants-12-00550-f002:**
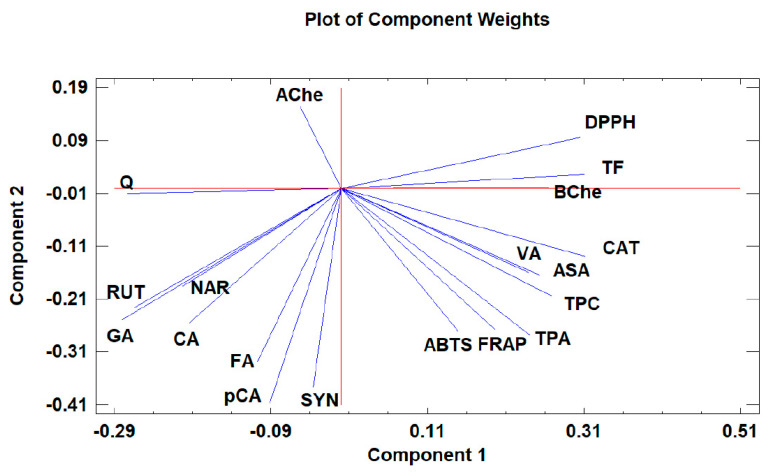
The loading plot of principal components 1 and 2 for the tested *Withania somnifera* samples extracted with two different protocols (hydromethanolic and aqueous extracts).

**Figure 3 antioxidants-12-00550-f003:**
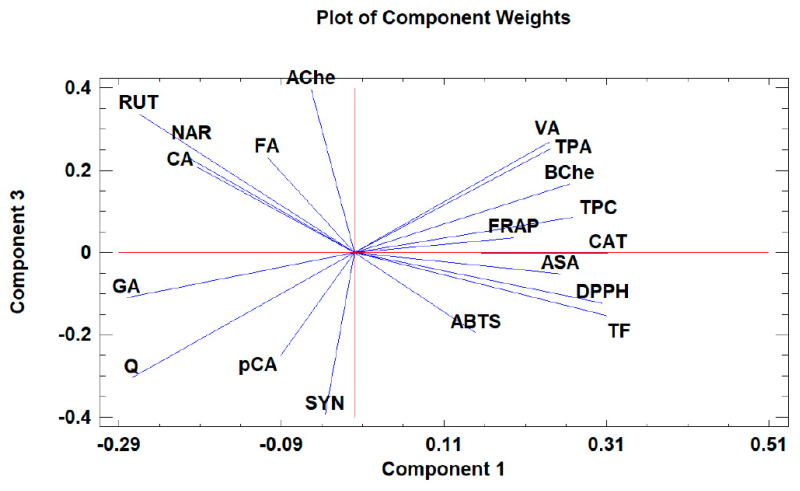
The loading plot of principal components 1 and 3 for the tested *Withania somnifera* samples extracted with two different protocols (hydromethanolic and aqueous extracts).

**Table 1 antioxidants-12-00550-t001:** List of analyzed commercial samples of *W. somnifera* L.

No.	Sample Name on the Package	Part of the Plant	Confection	Place of Origin of Plant Material	Place of Manufacture	Retailer Name
1.	Gold Ashwagandha	Root	Capsule	Unknown	Poland	Gemini
2.	Ashwagandha	Root	Tablet	India	Sweden	Gemini
3.	Ashwagandha	Root	Capsule	Unknown	USA	Gemini
4.	Ashwagandha	Root	Capsule	India	Germany	Fragaria
5.	Bicaps Ashwagandha	Root	Capsule	Unknown	Poland	Fragaria
6.	Ashvagandha	Rhizome	Capsule	India	India	Eko-Ziola
7.	Ashwagandha Root	Root	Solid powder	Unknown	Poland	Eko-Ziola
8.	Ashwagandha	Root	Capsule	Unknown	USA	Gemini
9.	Ashwagandha 100% Natural	Whole plant	Tablet	Unknown	Poland	Eko-Ziola
10.	Ashwagandha (root) in powder	Root	Ground root	Unknown	Poland	Gemini
11.	Ashwagandha	Root	Crushed root	Unknown	Poland	Nagietek
12.	Ashwagandha	Root	Sliced root	India	Poland	Auchan
13.	Ashwagandha	Root	Ground root	India	Poland	Auchan
14.	Ashwagandha	Root	Capsule	Unknown	USA	Gemini
15.	Bio Ashwagandha	Root	Ground root	India	Poland	Fragaria
16.	Ashwagandha	Root	Ground root	India	Poland	Nagietek
17.	Ashwagandha	Root	Capsule	India	Poland	Fragaria
18.	Ashwagandha	Root	Crushed root	India	Poland	Nagietek

**Table 2 antioxidants-12-00550-t002:** Validation parameters of the calibration curves for the compounds quantified (GA—gallic acid, CAT—catechin, VA—vanillic acid, CA—caffeic acid, pCA—p-coumaric acid, FA—ferulic acid, SYN—sinapinic acid, RUT—rutin, Q—quercetin, NAR—naringenin) in this study (*n* = 3).

Compounds	Regression Equation ^a^	Linearity (µg/mL)	R^2^	LODs (µg/mL)	LOQs (µg/mL)
GA	y = 1160x – 367,737	55.6–278	0.9872	2.5	8.4
CAT	y = 58,916x – 28,179	52.1–260.9	0.9971	3.5	13.2
VA	y = 25,058x + 109,038	54.8–274.2	0.9871	2.7	9.5
CA	y = 28,663x + 307,368	44.4–222.5	0.9985	3.4	12.5
*p*CA	y = 11,754x + 69,779	50.6–251.3	0.9956	3.1	12.0
FA	y = 21,142x – 309,335	55.1–265.4	0.9978	2.6	9.8
SYN	y = 19,097x − 296,645	57.2–286.5	0.9981	2.9	10.6
RUT	y = 20,735x + 44,710	47.6–238.0	0.9978	2.0	7.1
Q	y = 48,916x + 63,779	50.1–250.3	0.9983	3.1	9.7
NAR	y = 41,794x – 22,317	54.0–270.4	0.9934	2.6	9.1

^a^ y is the peak area; x refers to the concentration of compounds (µg/mL); LOD: limit of detection; LOQ: limit of quantification.

**Table 3 antioxidants-12-00550-t003:** Total phenolic compounds (TPC), flavonoids (TF), phenolic acids (TPA), L(+) ascorbic acid (ASA) contents, and DPPH, ABTS, and FRAP assays in the tested hydromethanolic and aqueous extracts of *W. somnifera* samples (mean ± SD).

	TPC	TF	TPA	ASA	DPPH	ABTS	FRAP
Hydromethanolic Extracts	mg GAE/g dw	mg QE/g dw	mg CAE/g dw	mg ASA/g dw	mg TE/100 g DW	mg g DW	µmol Fe^2+^/g dw
1	0.90 ± 0.12 ^e–g^*^†^	0.18 ± 0.07 ^fg^*	0.11 ± 0.09 ^h^*	7.82 ± 1.72 ^e–g^*	19.48 ± 2.06 ^e^*	16.65 ± 1.65 ^e^*	8.03 ± 0.09 ^e–g^*
2	2.03 ± 0.07 ^c^*	0.31 ± 0.02 ^c–f^*	2.04 ± 0.13 ^bc^*	13.32 ± 2.39 ^de^*	20.46 ± 2.^1 de6^*	26.75 ± 1.59 ^e^*	9.27 ± 0.69 ^ef^*
3	0.90 ± 0.01 ^e–g^*	0.21 ± 0.01 ^e–g^*	1.25 ± 0.10 ^de^*	6.77 ± 1.91 ^e–g^*	81.11 ± 2.47 ^a^*	47.10 ± 4.83 ^d^*	5.30 ± 0.32 ^gh^*
4	1.75 ± 0.28 ^cd^*	0.64 ± 0.04 ^a^*	1.68 ± 0.13 ^cd^*	22.23 ± 3.53 ^bc^*	42.53 ± 3.53 ^cd^*	27.29 ± 0.48 ^e^*	34.53 ± 2.48 ^b^
5	6.96 ± 0.04 ^a^*	0.26 ± 0.03 ^d–g^*	2.45 ± 0.10 ^ab^*	35.90 ± 1.10 ^a^	47.21 ± 5.66 ^c^*	122.54 ± 3.91 ^a^*	18.56 ± 1.54 ^c^*
6	0.65 ± 0.09 ^g^*	0.32 ± 0.01 ^c–e^*	2.67 ± 0.09 ^a^*	35.46 ± 0.96 ^a^*	32.71 ± 4.58 ^c–e^*	124.84 ± 6.67 ^a^*	19.30 ± 2.06 ^c^*
7	1.19 ± 0.03 ^d–g^*	0.40 ± 0.03 ^bc^*	0.54 ± 0.11 ^f–h^*	24.76 ± 2.84 ^b^*	22.16 ± 1.89 ^de^*	25.15 ± 6.41 ^e^*	10.70 ± 0.76 ^de^*
8	3.80 ± 0.61 ^b^*	0.48 ± 0.07 ^b^*	2.71 ± 0.81 ^a^*	17.06 ± 2.81 ^cd^*	52.96 ± 4.98 ^bc^*	99.92 ± 6.77 ^b^*	45.57 ± 0.13 ^a^*
9	0.82 ± 0.09 ^e–g^*	0.35 ± 0.04 ^cd^*	1.84 ± 0.39 ^c^*	20.98 ± 2.69 ^bc^*	38.36 ± 2.55 ^c–e^*	75.99 ± 9.86 ^c^	12.60 ± 0.30 ^d^*
10	0.80 ± 0.07 ^e–g^*	0.21 ± 0.02 ^e–g^*	0.32 ± 0.02 ^gh^*	6.99 ± 2.54 ^e–g^*	76.45 ± 1.39 ^a^*	23.47 ± 2.67 ^e^*	8.51 ± 0.91 ^ef^*
11	0.90 ± 0.22 ^e–g^*	0.22 ± 0.01 ^e–g^*	0.36 ± 0.04 ^gh^*	6.96 ± 0.80 ^e–g^*	80.82 ± 0.72 ^a^*	24.88 ± 0.49 ^e^*	8.29 ± 0.30 ^e–g^*
12	0.67 ± 0.03 ^fg^*	0.16 ± 0.01 ^g^*	0.30 ± 0.06 ^gh^*	11.19 ± 0.57 ^d–f^*	79.93 ± 2.05 ^a^*	17.04 ± 2.33 ^e^*	7.79 ± 0.89 ^e–g^*
13	0.65 ± 0.04 ^g^*	0.18 ± 0.04 ^fg^*	1.00 ± 0.05 ^ef^*	7.23 ± 0.82 ^e–g^*	80.34 ± 1.2 ^a^*	22.62 ± 6.26 ^e^*	13.02 ± 0.80 ^d^
14	1.33 ± 0.17 ^de^*	0.19 ± 0.03 ^fg^*	0.47 ± 0.05 ^gh^*	2.46 ± 0.23 ^g^*	85.75 ± 1.87 ^a^*	26.28 ± 2.76 ^e^*	8.20 ± 1.38 ^e–g^*
15	1.34 ± 0.07 ^de^*	0.16 ± 0.01 ^g^*	0.26 ± 0.01 ^gh^*	2.78 ± 0.43 ^g^*	87.82 ± 2.65 ^a^*	19.31 ± 3.46 ^e^*	10.01 ± 0.94 ^de^*
16	1.29 ± 0.12 ^d–f^*	0.18 ± 0.02 ^fg^*	0.30 ± 0.05 ^gh^*	3.73 ± 0.83 ^g^*	91.17 ± 0.12 ^a^*	27.28 ± 2.56 ^a^*	6.90 ± 0.38 ^f–h^*
17	1.35 ± 0.09 ^de^*	0.36 ± 0.09 ^cd^*	0.66 ± 0.02 ^fg^*	4.57 ± 1.09 ^fg^*	89.37 ± 2.43 ^a^*	21.68 ± 9.32 ^e^*	3.87 ± 0.35 ^h^*
18	1.57 ± 0.38 ^cd^*	0.19 ± 0.01 ^fg^*	0.42 ± 0.09 ^gh^*	3.86 ± 0.26 ^g^*	71.96 ± 4.55 ^ab^*	16.54 ± 0.88 ^e^*	4.55 ± 0.31 ^h^*
	**TPC**	**TF**	**TPA**	**ASA**	**DPPH**	**ABTS**	**FRAP**
**Aqueous extracts**	**mg GAE/g dw**	**mg QE/g dw**	**mg CAE/g dw**	**mg ASA/g dw**	**mg TE/100 g DW**	**mg g DW**	**µmol Fe^2+^/g dw**
1	3.46 ± 0.03 ^c–f^*	3.41 ± 0.33 ^ab^*	0.82 ± 0.13 ^h^*	36.92 ± 1.84 ^cd^*	281.92 ± 2.66 ^a^*	48.07 ± 3.36 ^e–g^*	16.86 ± 4.03 ^ef^*
2	5.19 ± 0.28 ^bc^*	2.29 ± 0.29 ^a–c^*	4.66 ± 0.83 ^de^*	60.42 ± 3.64 ^a^*	226.13 ± 4.36 ^bc^*	37.08 ± 3.43 ^f–h^*	35.54 ± 4.06 ^a–c^*
3	2.99 ± 0.76 ^d–f^*	2.47 ± 0.53 ^a–c^*	3.95 ± 0.99 ^ef^*	35.40 ± 3.31 ^c–e^*	174.76 ± 9.85 ^e–g^*	72.97 ± 3.66 ^bc^*	15.12 ± 2.27 ^ef^*
4	2.48 ± 0.08 ^ef^*	2.06 ± 0.28 ^bc^*	6.43 ± 0.35 ^bc^*	51.96 ± 5.02 ^ab^*	120.33 ± 5.06 ^hi^*	42.81 ± 0.91 ^e–g^*	34.08 ± 2.30 ^a–c^
5	5.06 ± 1.01 ^bc^*	2.14 ± 0.06 ^bc^*	7.10 ± 1.75 ^ab^*	34.11 ± 3.96 ^c–f^	212.66 ± 9.93 ^b–d^*	82.23 ± 5.23 ^a–c^*	9.31 ± 9.72 ^g^*
6	5.95 ± 1.97 ^a^*	3.60 ± 0.75 ^a^*	8.16 ± 0.84 ^a^*	19.54 ± 2.03 ^gh^*	177.04 ± 9.82 ^d–g^*	90.04 ± 3.26 ^a^*	13.87 ± 2.22 ^f^*
7	2.87 ± 0.34 ^d–f^*	1.62 ± 0.02 ^c^*	2.49 ± 0.41 ^fg^*	54.73 ± 5.02 ^ab^*	127.35 ± 3.46 ^h^*	54.76 ± 5.86 ^de^*	14.80 ± 2.63 ^ef^*
8	5.87 ± 0.59 ^a^*	2.60 ± 0.26 ^a–c^*	5.68 ± 1.83 ^cd^*	28.37 ± 2.33 ^d–g^*	93.66 ± 7.86 ^i^*	85.88 ± 3.70 ^ab^*	31.28 ± 6.15 ^a–d^*
9	4.69 ± 0.42 ^b–d^*	1.62 ± 0.12 ^c^*	7.67 ± 0.35 ^ab^*	45.85 ± 4.36 ^bc^*	189.75 ± 6.72 ^d–f^*	72.71 ± 4.32 ^bc^	40.01 ± 3.14 ^a^*
10	3.67 ± 0.08 ^c–e^*	2.36 ± 0.26 ^a–c^*	1.20 ± 0.26 ^gh^*	23.26 ± 2.53 ^d–h^*	212.18 ± 8.04 ^b–e^*	85.03 ± 3.13 ^ab^*	24.33 ± 1.14 ^c–f^*
11	3.29 ± 0.01 ^d–f^*	2.37 ± 0.20 ^a–c^*	1.37 ± 0.22 ^gh^*	23.00 ± 2.57 ^e–h^*	204.31 ± 9.37 ^c–e^*	40.47 ± 7.59 ^e–g^*	18.71 ± 0.78 ^ef^*
12	3.08 ± 0.18 ^d–f^*	2.20 ± 1.10 ^a–c^*	0.82 ± 0.28 ^h^*	21.32 ± 3.03 ^f–h^*	154.50 ± 4.36 ^f–h^*	21.31 ± 5.93 ^h^*	21.31 ± 0.66 ^d–f^*
13	2.57 ± 0.14 ^ef^*	2.71 ± 0.05 ^a–c^*	1.43 ± 0.28 ^gh^*	29.54 ± 2.07 ^d–g^*	154.67 ± 6.62 ^f–h^*	26.21 ± 1.57 ^h^*	13.17 ± 0.73 ^f^
14	4.52 ± 0.38 ^b–d^*	2.51 ± 0.40 ^a–c^*	2.62 ± 0.02 ^fg^*	34.57 ± 1.59 ^c–f^*	244.23 ± 3.93 ^b^*	42.40 ± 4.19 ^e–g^*	21.77 ± 4.60 ^d–f^*
15	3.48 ± 0.31 ^c–f^*	1.93 ± 0.37 ^c^*	1.57 ± 0.27 ^gh^*	27.79 ± 5.65 ^d–g^*	200.32 ± 8.06 ^c–e^*	31.58 ± 2.53 ^gh^*	26.42 ± 5.81 ^b–e^*
16	2.24 ± 0.18 ^ef^*	2.21 ± 0.47 ^a–c^*	0.99 ± 0.26 ^h^*	13.97 ± 3.10 ^h^*	192.49 ± 9.02 ^c–e^*	66.03 ± 5.09 ^cd^*	18.97 ± 2.91 ^ef^*
17	3.12 ± 0.07 ^d–f^*	1.44 ± 0.22 ^c^*	5.21 ± 0.37 ^c–e^*	21.46 ± 2.30 ^f–h^*	143.26 ±5.48 ^g–h^*	75.04 ± 5.56 ^a–c^*	37.61 ± 1.36 ^ab^*
18	1.73 ± 0.23 ^f^*	2.43 ± 0.90 ^a–c^	1.44 ± 0.25 ^gh^*	28.96 ± 3.39 ^d–g^*	118.91 ± 7.38 ^hi^*	52.44 ± 5.32 ^d–f^*	18.14 ± 3.56 ^ef^*

^†^ Means followed by the same letter within the same column and for the same extraction method indicate no significant difference among samples according to Tukey’s HSD test (*p* < 0.05). The asterisk (*) symbol indicates significant differences between the different extraction protocols for the same sample according to Student’s *t* test (*p* < 0.05). Arabic numbers in the first column denote commercial samples of *W. somnifera* listed in Table 1.

**Table 4 antioxidants-12-00550-t004:** Phenolic compounds in *W. somnifera* L. samples (mean ± standard deviation).

	GA	CAT	VA	CA	FA	SYN	*p*CA	RUT	Q	NAR
Hydromethanolic Extracts										
Sample No	(µg/g)	(mg/g)	(µg/g)	(µg/g)	(µg/g)	(µg/g)	(µg/g)	(µg/g)	(mg/g)	(µg/g)
1	136.50 ± 2.81 ^c†^	1.04 ± 0.23 ^d^*	106.93 ± 1.32 ^b^*	305.28 ± 2.43 ^b^	262.55 ± 3.54 ^a–c^	ND	ND	1085.06 ± 5.35 ^a^	1.25 ± 0.03 ^bc^*	573.38 ± 2.43 ^a^
2	252.71 ± 3.87 ^b^*	2.37 ± 0.76 ^b^*	ND	367.33 ± 3.64 ^b^*	479.74 ± 2.57 ^a^*	ND	271.91 ± 2.45 ^c^	764.42 ± 3.52 ^a–c^*	1.26 ± 0.04 ^b^*	499.58 ± 3.82 ^a^*
3	125.78 ± 1.36 ^c^*	1.42 ± 0.67 bc*	99.08 ± 2.03 ^b^*	295.96 ± 2.62 ^b^*	270.47 ± 2.53 ^a–c^	ND	ND	414.31 ± 4.75 ^c–e^	1.22 ± 0.06 ^bc^*	47.76 ± 3.89 ^c–e^
4	288.22 ± 2.71 ^b^*	6.54 ± 0.32 ^a^*	ND	360.82 ± 1.75 ^b^*	458.57 ± 3.75 ^ab^*	132.56 ± 2.32 ^b^	277.71 ± 2.63 ^b^*	747.88 ± 3.86 ^a–d^*	1.49 ± 0.03 ^b^*	316.44 ± 2.81 ^b^*
5	120.67 ± 1.18 ^c^*	1.13 ± 0.12 ^d^*	150.46 ± 4.21 ^a^*	331.60 ± 3.67 ^b^*	ND	185.43 ± 3.72 ^a^*	241.38 ± 1.34 ^d^	915.71 ± 4.63 ^ab^*	0.85 ± 0.03 ^c^	496.81 ± 3.87 ^a^
6	428.01 ± 2.82 ^a^*	2.16 ± 0.76 ^d^*	ND	334.70 ± 3.32 ^b^*	174.93 ± 1.53 ^c^*	134.11 ± 2.98 ^b^	466.77 ± 3.78 ^a^	431.65 ± 3.57 ^c–e^*	1.28 ± 0.07 ^b^*	69.30 ± 3.82 ^c–e^*
7	119.91 ± 2.57 ^c^	2.28 ± 0.57 ^b^*	105.64 ± 1.32 ^b^	312.01 ± 4.04 ^b^	ND	ND	151.29 ± 1.83 ^e^	423.91 ± 4.24 ^c–e^	1.37 ± 0.06 ^b^	49.22 ± 2.34 ^c–e^
8	118.23 ± 1.52 ^c^*	5.33 ± 1.67 ^d^*	ND	505.28 ± 2.52 ^a^*	238.63 ± 2.56 ^bc^*	87.89 ± 1.65 ^c^	278.58 ± 2.43 ^b^	594.41 ± 5.75 ^b–e^	2.16 ± 0.12 ^a^*	174.01 ± 3.89 ^c^*
9	115.02 ± 2.56 ^c^	0.75 ± 0.12 ^d^*	104.64 ± 2.43 ^b^*	313.55 ± 2.45 ^b^*	ND	ND	ND	373.55 ± 2.54 ^de^	1.28 ± 0.07 ^b^*	137.85 ± 2.79 ^cd^*
10	115.85 ± 1.29 ^c^*	0.95 ± 0.25 ^d^*	ND	310.57 ± 3.89 ^b^	ND	ND	ND	421.90 ± 2.87 ^c–e^*	1.37 ± 0.05 ^b^*	39.17 ± 1.93 ^de^*
11	123.65 ± 2.82 ^c^*	1.14 ± 0.57 ^d^*	ND	327.03 ± 3.49 ^b^	262.11 ± 2.54 ^a–c^*	ND	ND	377.81 ± 2.48 ^de^*	1.46 ± 0.07 ^b^*	ND
12	114.17 ± 3.16 ^c^	1.05 ± 0.52 ^d^*	100.90 ± 2.73 ^b^	305.86 ± 2.10 ^b^	ND	ND	ND	403.12 ± 3.75 ^c–e^	1.45 ± 0.03 ^b^*	26.66 ± 2.79 ^de^
13	117.79 ± 1.27 ^c^	0.95 ± 0.63 ^d^*	ND	309.70 ± 3.15 ^b^	255.37 ± 3.83 ^bc^	ND	ND	381.22 ± 3.45 ^de^	1.32 ± 0.08 ^b^*	ND
14	120.69 ± 2.18 ^c^	1.06 ± 0.29 ^d^*	101.83 ± 1.83 ^b^*	309.44 ± 2.07 ^b^	256.65 ± 3.82 ^bc^	ND	ND	389.54 ± 4.01 ^c–e^	1.25 ± 0.06 ^b^*	ND
15	111.67 ± 2.62 ^c^*	0.81 ± 0.26 ^d^*	ND	300.01 ± 2.19 ^b^	244.50 ± 2.96 ^bc^	ND	ND	743.20 ± 4.24 ^a–d^	1.32 ± 0.04 ^b^*	443.46 ± 3.27 ^ab^*
16	122.77 ± 2.17 ^c^	1.06 ± 0.17 ^d^*	ND	297.99 ± 2.92 ^b^	266.68 ± 3.82 ^a–c^	ND	ND	391.45 ± 3.29 ^c–e^	1.40 ± 0.06 ^b^*	ND
17	114.70 ± 1.72 ^c^	0.75 ± 0.52 ^d^*	101.63 ± 2.10 ^b^*	328.55 ± 3.56 ^b^	ND	ND	ND	361.81 ± 2.08 ^e^*	1.23 ± 0.07 ^bc^	ND
18	108.92 ± 1.96 ^c^	0.84 ± 0.13 ^d^*	ND	297.64 ± 2.91 ^b^	ND	ND	ND	360.34 ± 1.92 ^e^	1.26 ± 0.04 ^b^*	ND
	**GA**	**CAT**	**VA**	**CA**	**FA**	**SYN**	***p*CA**	**RUT**	**Q**	**NAR**
**Aqueous extracts**										
**Sample no**	**(mg/g)**	**(mg/g)**	**(µg/g)**	**(µg/g)**	**(µg/g)**	**(µg/g)**	**(µg/g)**	**(µg/g)**	**(mg/g)**	**(mg/g)**
1	ND	6.66 ± 1.02 ^ef^*	705.71 ± 1.78 ^d^*	ND	ND	95.07 ± 1.04 ^a^	ND	ND	1.06 ± 0.03 ^ab^*	ND
2	0.92 ± 0.05 ^cd^*	9.44 ± 1.87 ^bc^*	986.42 ± 2.87 ^a^	633.50 ± 1.32 ^a^*	264.77 ± 2.19 ^a^*	34.45 ± 0.43 ^c^	58.18 ± 1.43 ^d^	337.27 ± 2.03 ^b^*	0.65 ± 0.05 ^ab^*	1.60 ± 0.21 ^c^*
3	0.95 ± 0.04 ^b–d^*	7.04 ± 0.84 ^d–f^*	674.14 ± 2.56 ^e^*	377.88 ± 1.92 ^b^*	ND	ND	109.28 ± 1.53 ^a^	ND	0.68 ± 0.04 ^ab^*	ND
4	1.95 ± 0.08 ^a^*	8.71 ± 0.68 ^b–d^*	103.96 ± 1.48 ^h^	340.13 ± 1.78 ^b^*	297.31 ± 1.88 ^a^*	ND	89.27 ± 1.93 ^b^*	105.23 ± 1.73 ^e^*	0.37 ± 0.03 ^bc^*	0.91 ± 0.06 ^d^*
5	1.04 ± 0.05 ^b^*	11.93 ± 1.79 ^a^*	736.68 ± 2.77 ^c^*	102.93 ± 1.83 ^d^*	186.44 ± 1.76 ^b^	41.04 ± 0.54 ^b^*	25.09 ± 0.43 ^f^	115.52 ± 2.94 ^e^*	ND	ND
6	0.92 ± 0.04 ^cd^*	7.69 ± 2.75 ^c–f^*	884.32 ± 1.82 ^b^	273.31 ± 1.89 ^c^*	277.94 ± 2.01 ^a^*	ND	15.22 ± 0.69 ^g^	357.07 ± 4.92 ^ab^*	0.68 ± 0.03 ^ab^*	1.67 ± 0.04 ^c^*
7	ND	7.70 ± 1.88 ^c–f^*	ND	ND	ND	ND	62.23 ± 0.88 ^d^	ND	ND	ND
8	1.01 ± 0.03 ^bc^*	10.50 ± 2.96 ^ab^*	142.42 ± 1.78 ^g^	129.09 ± 2.04 ^e^*	291.33 ± 1.89 ^a^*	ND	50.12 ± 0.72 ^e^	199.28 ± 2.12 ^d^*	0.72 ± 0.05 ^ab^*	1.77 ± 0.08 ^c^*
9	ND	8.12 ± 2.03 ^c–e^*	874.62 ± 3.52 ^b^*	245.02 ± 2.12 ^c^*	204.39 ± 2.97 ^b^	ND	23.43 ± 0.65 ^f^	376.21 ± 1.93 ^a^*	0.33 ± 0.01 ^c^*	0.81 ± 0.03 ^d^
10	0.95 ± 0.06 ^b–d^*	6.48 ± 0.78 ^ef^*	ND	ND	ND	ND	ND	ND	0.99 ± 0.02 ^ab^*	2.41 ± 0.44 ^b^*
11	0.88 ± 0.04 ^d^*	6.78 ± 0.52 ^ef^*	ND	ND	271.60 ± 1.67 ^a^*	ND	ND	ND	1.03 ± 0.05 ^ab^*	ND
12	ND	6.48 ± 0.96 ^ef^*	ND	ND	ND	ND	ND	ND	0.74 ± 0.04 ^ab^*	ND
13	ND	6.42 ± 0.99 ^ef^*	ND	ND	ND	ND	ND	ND	1.02 ± 0.03 ^ab^*	ND
14	ND	6.66 ± 0.76 ^ef^*	710.96 ± 2.26 ^d^*	ND	ND	ND	76.11 ± 1.02 ^c^	ND	1.07 ± 0.04 ^a^*	ND
15	0.87 ± 0.05 ^d^*	6.34 ± 0.77 ^ef^*	ND	ND	ND	ND	ND	ND	0.82 ± 0.05 ^ab^*	2.49 ± 0.12 ^b^*
16	ND	6.53 ± 0.85 ^ef^*	ND	ND	ND	ND	ND	ND	1.06 ± 0.03 ^ab^*	ND
17	ND	7.64 ± 1.74 ^c–f^*	631.10 ± 2.78 ^f^*	ND	ND	ND	73.07 ± 1.65 ^c^	254.44 ± 1.79 ^c^*	ND	ND
18	ND	5.98 ± 0.23 ^f^*	ND	ND	ND	ND	ND	ND	0.99 ± 0.02 ^ab^*	9.92 ± 0.21 ^a^

^†^ Means followed by the same letter within the same column and for the same extraction method indicate no significant difference among samples according to Tukey’s HSD test (*p* < 0.05). The asterisk (*) symbol indicates significant differences between the different extraction protocols for the same sample according to Student’s *t* test (*p* < 0.05). ND—not detected; GA—gallic 331 acid, CAT—catechin, VA—vanillic acid, CA—caffeic acid, pCA—p-coumaric acid, FA—ferulic acid, SYN—sinapinic acid, RUT—rutin, Q—quercetin, NAR—naringenin.

**Table 5 antioxidants-12-00550-t005:** Antibacterial activity of the aqueous extracts from *W. somnifera* (mg/mL) expressed as MIC and MBC.

	*S. aureus* ATCC 6538		MRSA 6374		MRSA N315		MRSA 18582		MRSA 12673		*S. epidermidis* ATCC 14990	
Sample No	MIC	MBC	MIC	MBC	MIC	MBC	MIC	MBC	MIC	MBC	MIC	MBC
1.	32	>32	8	32	>32	>32	16	16	>32	>32	16	>32
2.	2	8	0.5	1	4	32	8	16	32	>32	4	32
3.	16	32	8	32	>32	>32	>32	>32	>32	>32	>32	>32
4.	1	16	4	>32	2	4	2	8	4	>32	4	32
6.	2	8	8	>32	4	8	4	16	8	>32	4	16
8.	4	16	16	>32	4	8	4	8	32	>32	4	16
9.	2	8	1	4	2	32	4	16	8	>32	4	16
10.	16	>32	1	>32	4	>32	4	>32	2	>32	4	8
11.	16	>32	8	>32	8	>32	4	>32	16	>32	4	32
15.	4	32	4	>32	2	>32	8	>32	4	>32	2	8
	** *S. pneumoniae* **		** *S. pyogenes* **		** *E. hirae* **		** *E. faecalis* **		***B.subtillis* ATCC 6633**		** *C. diphtheriae* **	
**Sample No**	**MIC**	**MBC**	**MIC**	**MBC**	**MIC**	**MBC**	**MIC**	**MBC**	**MIC**	**MBC**	**MIC**	**MBC**
1.	16	32	0.25	2	32	>32	32	>32	2	4	0.5	1
2.	8	16	0.5	1	32	>32	8	>32	4	4	1	0.5
3.	>32	>32	0.25	>32	>32	>32	>32	>32	>32	>32	0.25	>32
4.	4	8	0.25	0.25	8	>32	4	8	0.5	1	0.1	0.25
6.	4	8	0.25	0.5	16	>32	4	>32	1	2	1	1
8.	8	16	4	>32	>32	>32	8	>32	4	4	8	8
9.	4	8	4	>32	16	>32	1	>32	8	8	>32	>32
10.	4	8	4	>32	16	>32	4	>32	1	1	0.5	4
11.	4	8	4	>32	32	>32	4	>32	1	1	>32	>32
15.	2	8	4	>32	8	>32	1	>32	0.5	0.5	>32	>32

MIC: minimum inhibitory concentration; MBC: minimum bactericidal concentration.

**Table 6 antioxidants-12-00550-t006:** Acetylcholinesterase (AChE) and butyrylcholinesterase (BChE) activity (%) compared to control samples (without extract) of hydromethanolic and aqueous extracts of *W. somnifera* commercial samples.

Cholinesterase Inhibition				
	AChE		BChE	
Sample No.	Hydromethanolic Extracts	Aqueous Extracts	Hydromethanolic Extracts	Aqueous Extracts
1	83.24 ± 2.63 ^d^*†	71.25 ± 0.59 ^e^*	54.30 ± 1.76 ^bc^*	101.35 ± 0.85 ^c^*
2	64.36 ± 5.27 ^i^*	46.46 ± 2.33 ^k^*	53.60 ± 1.37 ^c^*	90.17 ± 3.94 ^f^*
3	63.83 ± 0.75 ^i^*	53.33 ± 5.3 ^i^*	40.07 ± 2.74 ^e^*	94.29 ± 4.53 ^e^*
4	40.69 ± 9.4 ^k^*	59.17 ± 2.37 ^g^*	21.24 ± 0.83 ^h^*	60.30 ± 2.07 ^l^*
5	123 ± 11.77 ^a^*	46.46 ± 5.6 ^k^*	36.79 ± 1.89 ^f^*	76.44 ± 3.54 ^g^*
6	84.04 ± 8.27 ^d^*	57.29 ± 3.86 ^h^*	55.96 ± 3.42 ^b^*	105.71 ± 4.39 ^b^*
7	107.45 ± 5.34 ^b^	106.88 ± 5.04 ^b^	78.28 ± 2.15 ^a^*	138.27 ± 1.12 ^a^*
8	21.28 ± 1.04 ^l^*	67.29 ± 3.26 ^f^*	18.55 ± 4.15 ^i^*	69.94 ± 4.85 ^i^*
9	77.93 ± 0.38 ^e^*	109.58 ± 3.54 ^a^*	21.41 ± 1.42 ^h^*	75.68 ± 1.31 ^gh^*
10	47.07 ± 0.58 ^j^*	41.88 ± 2.06 ^n^*	16.30 ± 1.17 ^j^*	49.95 ± 3.74 ^n^*
11	65.96 ± 1.50 ^i^*	43.96 ± 2.06 ^l^*	35.64 ± 0.21 ^f^*	74.91 ± 5.05 ^h^*
12	75.00 ± 4.51 ^f^*	51.46 ± 2.67 ^j^*	41.07 ± 0.75 ^e^*	69.43 ± 1.04 ^i^*
13	68.88 ± 1.88 ^h^*	58.96 ± 1.31 ^g^*	17.82 ± 2.83 ^ij^*	33.53 ± 0.85 ^o^*
14	71.81 ± 0.75 ^g^*	36.46 ± 3.26 ^o^*	46.42 ± 1.17 ^d^*	99.35 ± 1.31 ^d^*
15	83.24 ± 0.38 ^d^*	42.92 ± 2.43 ^m^*	46.15 ± 0.78 ^d^*	58.07 ± 3.91 ^m^*
16	89.89 ± 3.01 ^c^	88.13 ± 4.45 ^c^	27.11 ± 2.39 ^g^*	61.64 ± 9.51 ^k^*
17	120.74 ± 1.53 ^a^*	106.46 ± 2.45 ^b^*	47.81 ± 1.86 ^d^*	89.19 ± 1.51 ^f^*
18	87.50 ± 0.38 ^c^*	77.5 ± 4.73 ^d^*	53.64 ± 3.96 ^c^*	64.42 ± 1.38 ^j^*

^†^ Means followed by the same letter within the same column and for the same extraction method indicate no significant difference among samples according to Tukey’s HSD test (*p* < 0.05). The asterisk (*) symbol indicates significant differences between the different extraction protocols for the same sample according to Student’s *t* test (*p* < 0.05).

## Data Availability

Data is available upon request.

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
