# Peer review of "Withania somnifera L.: Phenolic Compounds Composition and Biological Activity of Commercial Samples and Its Aqueous and Hydromethanolic Extracts"

_antioxidants, 2023, doi:10.3390/antiox12030550_

Round 1
Reviewer 1 Report
The paper under review in the present version needs some improvements, before recommending its publication. All my suggestions are reported below:
-The introduction section describes the state of the art well, citing a proper number of published papers. In the light of the above, the authors should highlight more the novelty of their work, which now appears quite poor.
-Ref 8 at line 79 was not properly cited. Indeed, the article 10.1016/j.jpba.2016.07.032 refers in its introduction section to a previous one for anticancer activity, while it reported the development of a UHPLC-MS method for the quantification of six compounds. The same can be observed for refs 9 and 13.
-The use as tea infusion is not enough to justify the decision to test only aqueous extracts for antibacterial activity. Any extract could be evaluated, even if it does not derive from infusion.
-In my opinion, the evaluation of bioactive properties testing 0.1 mL of the extracts without specifying the concentration is incorrect. Moreover, a good comparison with literature data should include the IC50 value, which can be calculated only by testing increasing doses of the sample under study.
-Results of colorimetric assays (e.g. TPC, TF and TPA) are not in line with the chemical composition analysis. A reason could be found in the presence of other metabolites herein not investigated, due to the lack of standard compounds. It is strongly recommended to evaluate the chemical composition by LC/MS tools in order to have a more comprehensive view.
-Line 349: please check the statement compared to the table. Sample n. 5 was not the richest in TF and TPA.
-Line 359-361: the sentence should be deepened by comparison with cited literature.
-It is well-known that TPC values could be compromised by the presence of other compounds that are not phenols (thus, overestimated). I would have expected lower TPA values in every sample. Could the authors explain why in some cases (e.g. HM n. 3, 9 ,13 or aqueous 4-6) the obtained values are higher?
-Is the antibacterial activity correlated to the chemical composition? A brief discussion should be added
Other issues:
-the abstract section is too long (360 words instead of a maximum of 200, as suggested by the guide for authors). It contains too general information. Without reading the whole manuscript, "samples n. 2, 4 or 6" have poor significance.
-DPPH assay acronym should not contain the word "hydrate"
-Please, add acronyms in the caption of Table 4
-Some typos need correction.
Reviewer 2 Report
The present study was aimed at the evaluation of phenolic compounds composition and biological activity of Withania somnifera L. commercial samples. The present paper is valuable, just some improvements would be useful.
- I would recommend not using sample numbers both in the abstract and conclusions. Maybe it is possible to rename that it would be more informative, for example, a sample from India, capsules from a Poland producer, etc.
- line 48 - should be "poison gooseberry"
- line 195 - add the volume and concentration of the sample.
- lines 199-202 - methodology is not very clear: specify volume and concentration of the sample, add concentrations of Arnov's reagent, HCl, NaOH.
- How many repetitions were performed for antibacterial analysis? What was used as a positive and negative control?
- Fig. 2 and 3 - should be points for decimal separation.
Line 431 - Latin names must be in italics.
Reviewer 3 Report
Dear Editor,
I reviewed the manuscript antioxidants-2204029, "Withania Somnifera L.: Phenolic Compounds Composition and 2 Biological Activity of Commercial Samples and its Aqueous and Hydromethanolic Extracts," which was submitted for publication in Antioxidants.
The presented manuscript is an experimental study focused on the comparison of the phenolic composition (chromatographic and photometric analysis methods) of commercial samples of W. somnifera and the evaluation of antioxidant, antibacterial, acetylcholinesterase, and butyrylcholinesterase inhibitory activities.
In my opinion, this is a good experimental work that will be interesting to experts in plant science and can be published in Antioxidants after minor revision.
My comments are related to typos and format:
- Table 3.
- Check the format of the column names. Use bold text.
- Apply the subscript to the first value in the Table.
- Remove the Bold text for lines 423-426.
- Table 4.
It's probably best to divide the table into two parts.
- The format of the column names can be optimized.
- Figures 1 and 2.
- Please use a decimal point instead of a decimal comma.
- Supplement Table 1.
- Please use a decimal point instead of a decimal comma.
- Please remove "%" from the Table.
- Supplement Tables 2 and 3.
- Please add the description of abbreviations in Tables 2 and 3.
Round 2
Reviewer 1 Report
The authors took into consideration most of my suggestions, modifying the manuscript accordingly. The manuscript has been improved. I am still not very convinced about the lack of a comprehensive chemical characterization, which could support biological activity data. Indeed, new important insights into the contribution of phenolic compounds in bioactive properties could not exclude an untargeted approach in identifying bioactive molecules, other than those compared to commercial standards.